# Neuroprotective Effects of Choline and Other Methyl Donors

**DOI:** 10.3390/nu11122995

**Published:** 2019-12-06

**Authors:** Rola A. Bekdash

**Affiliations:** Department of Biological Sciences, Rutgers University, Newark, NJ 07102, USA; rbekdash@newark.rutgers.edu

**Keywords:** brain, choline, dementia, epigenetics, methyl-donors, nutrition, neuroprotection

## Abstract

Recent evidence suggests that physical and mental health are influenced by an intricate interaction between genes and environment. Environmental factors have been shown to modulate neuronal gene expression and function by epigenetic mechanisms. Exposure to these factors including nutrients during sensitive periods of life could program brain development and have long-lasting effects on mental health. Studies have shown that early nutritional intervention that includes methyl-donors improves cognitive functions throughout life. Choline is a micronutrient and a methyl donor that is required for normal brain growth and development. It plays a pivotal role in maintaining structural and functional integrity of cellular membranes. It also regulates cholinergic signaling in the brain via the synthesis of acetylcholine. Via its metabolites, it participates in pathways that regulate methylation of genes related to memory and cognitive functions at different stages of development. Choline-related functions have been dysregulated in some neurodegenerative diseases suggesting choline role in influencing mental health across the lifespan.

## 1. Introduction

Emerging evidence suggests that environmental factors such as repeated exposure to drugs of abuse, stressors, lifestyle factors such as physical activity and diet have substantial effects on mental health and well-being. These factors interact with genes and modulate their expression and function by epigenetic mechanisms [1]. Dysregulation of gene expression by changes in gene methylation has been demonstrated in neurological disorders [2,3], and reported during aging and in some neurodegenerative diseases [4]. It has been suggested that an epigenetic-based approach for the treatment of these disorders is a promising avenue to undertake since the components of the epigenetic machinery have been successfully used in preclinical trials for the management of many diseases including neurodegenerative diseases [5].

The role of the micronutrient choline and other methyl-donors in normal fetal growth and development has been well studied [6,7]. Recently, choline emerged as an epigenetic modifier of the genome that modulates neuronal gene expression and alters brain function by affecting the availability of S-adenosylmethionine (SAM) [8,9], a main cellular methyl-donor for key epigenetic mechanisms such as DNA and histone methylation. SAM synthesis is dependent on the availability of dietary components or nutrients such as folate, VitB6, VitB12, methionine, choline and betaine [10]. SAM also has been shown to act as an enhancer of cognitive performance and ability [11]. This poses the question whether these nutrients that are classified as methyl donors could improve cognitive functions with aging and in some neurodegenerative diseases like Alzheimer’s disease. Identification and implementation of effective nutritional strategies early in life may then optimize cognitive functions and mental health throughout life [12].

Choline is not only an essential micronutrient needed for proper organs’ functions but also a neuroprotectant needed for normal development and growth of fetal brain [13,14]. Besides its pivotal role in normal functioning of the developing brain [15], supplementation of choline may mitigate age-related memory decline later in life [16]. Animal and human studies have shown that prenatal or perinatal supplementation of choline influences fetus long-term health and its vulnerability to diseases later on in life. For example, prenatal or perinatal supplementation of choline in rodents attenuated stress, modulated behavior [17,18] and improved memory and cognitive functions [19,20] in the adult offspring. Supplementation of choline and other methyl-donors such as VitB6, VitB12, folate and methionine may attenuate age-related cognitive decline with aging [21,22]. Choline has also been shown to mitigate symptoms associated with genetically-related neurodevelopmental disorders such as Down Syndrome [23,24,25] and Rett Syndrome [26] and neurodegenerative diseases such as Alzheimer’s disease (AD) [27]. These various effects of choline on the functioning of the developing and aging brain often correlated with epigenetic changes such as histone marks changes or DNA methylation changes of key genes related to cognitive functions [28]. This suggests that choline as an environmental factor and as a micronutrient has neuroprotective effects and programs brain development during early life via an epigenetic mechanism of action [29,30].

In this review, we will focus on the role of the micronutrient choline as a neuroprotectant and summarize research findings that demonstrated a link between maternal supplementation of dietary choline and improvement of cognitive abilities during early development. We will also summarize findings that suggested the potential use of this micronutrient in combination with other methyl-donors to delay or prevent decline in cognitive functions in the aging brain and in some neurodegenerative diseases such as Alzheimer’s disease. At the end of this review, we will address the adverse effects of inadequate supplementation of methyl-donors on neurodevelopment and cognitive functions.

## 2. The Physiological Functions of Choline and Other Methyl Donors

The Food and Nutrition Board of the Institute of the National Academy of Sciences of the United States recommends an adequate intake (AI) of 7.5 mg of choline daily per kg of body weight in humans [31]. The American Academy of Pediatrics identified several nutrients as essential for brain growth and development including choline, VitB6, and VitB12. The US Food and Drug Administration recommended 7 mg of choline per 100 Kcal as an optimal value for infants [32]. Despite this recommendation, some individuals do not meet their daily choline requirement and other individuals increase their intake above the recommended value. AI varies from one individual to another and this variation is influenced by several factors such as age, sex, genetic and environmental factors [33,34,35]. For example, an elevated intake of choline is required for pregnant women to support fetal growth and development and improve neurocognitive functions of offspring later in life [12,32]. Premenopausal women require less intake of choline compared to postmenopausal women. This is explained by the fact that estrogen induces phosphotidylethanolamine-N-methyltransferase (PEMT) expression in women by binding to its estrogen responsive element (ERE). PEMT catalyzes the reaction for de novo synthesis of choline [36]. Decline of estrogen levels after menopause or the presence of genetic polymorphisms (SNPs) in the hepatic PEMT gene predisposes women to a depletion in choline as they age and increase their susceptibility to diseases [37]. Depletion of choline during adulthood has also been linked to organ or cellular dysfunction such as nonalcoholic fatty liver, liver damage, muscle damage, lymphocyte apoptosis and DNA damage [38,39,40].

Choline and other methyl-donors have prominent role at the level of the brain. Several studies conducted in humans and animal models demonstrated that depletion or supplementation of choline during sensitive periods of brain development such as prenatal, perinatal or early postnatal life contributes to the etiology of neural tube defects [41,42]. The increased risk of neural tube defects has been linked to genetic variant in 5,10 methylenetetrahydrofolate dehydrogenase (MTHFD1) allele (rs2236225), an enzyme that plays a role in folate metabolism [43]. Increasing maternal intake of choline during the third trimester of pregnancy has been linked to a decrease in placental expression of the antiangiogenic factor Fms-like tyrosine kinase-1 suggesting the potential role of choline in reducing preeclampsia in pregnant women [44]. In the context of stress, the same research group showed that maternal choline supplementation altered the methylation of the promoter of some stress-related genes including corticotropin-releasing hormone gene (Crh) and glucocorticoid gene (Nr3c1) in the placenta and cord blood. These molecular changes correlated with a decrease in the levels of the stress hormone cortisol in the cord blood [45]. Changes in choline levels have also been associated with alteration in cognitive functions with aging [46,47], Alzheimer’s disease (AD) [48,49], behavioral changes in animal model of prenatal alcohol exposure [17,50,51,52], and with symptoms of neurodevelopmental disorders such as Rett Syndrome [26,53] and Down Syndrome [25,54,55].

Choline is classified as an essential micronutrient by the Food and Nutrition Board (FNB) of the Institute of Medicine of the National Academy of Sciences. It is naturally derived from wide variety of foods such as eggs, beans, fish, nuts, seeds, whole grain, some meats and vegetables from which we obtain our needs [34]. The endogenous synthesis of choline occurs mainly in the liver and in other tissues of the body as well including the brain. The de novo synthesis of choline is catalyzed by the enzymatic activity of PEMT via the sequential methylation of phosphotidylethanolamine (PE) using SAM as a methyl donor [56]. The oxidation of choline into its metabolite betaine by choline oxidase in the liver plays a role in the formation of methionine and SAM, methyl-donors that are used in methylation pathways. SAM is considered the main methyl-donor for DNA methyltransferases (DNMTs) and histone methyltransferases (HMTs), key enzymes that catalyze DNA methylation and histone methylation of genes (Figure 1). A portion of choline is acetylated into acetylcholine (Ach) by the enzymatic activity of choline acetyltransferase (ChAT). The de novo synthesis of choline mentioned earlier is not sufficient to meet human’s daily requirement of this micronutrient. Thus, it is essential to obtain it from the diet or take it as a supplement following the recommended daily intake for humans across all ages. Supplementation of choline in terms of dosage, and frequency of intake and the population involved should be evaluated and determined by health professionals.

The levels of free choline and its metabolites increase in the brain of humans and rodents after ingestion [57,58], suggesting the transport of this hydrophilic molecule to the brain via specific transporters. The brain cannot synthesize choline but gets its need from three sources: (1) free choline that crosses the blood brain barrier via choline transporters after which choline will be stored in neuronal membranes as phosphotidylcholine (PC) for Ach biosynthesis, (2) Ach degradation by Acetylcholine Esterase (AchE) and (3) hydrolysis of PC by the activity of phosphotidylcholine-specific phospholipases [59]. The enzymatic activities of ChAT and PEMT were detected in some neuronal terminals suggesting that neurons have the ability to synthesize choline and use it for Ach biosynthesis [60,61,62,63].

Choline is the precursor for several membrane phospholipids such as PC, PE and sphingomyelin (SM). These choline-containing phospholipids maintain structural and functional integrity of cellular membranes including neuronal membranes. They also participate in other physiological processes essential for normal brain development and functioning including cellular signaling, neuronal myelination and division [64,65,66]. Choline as a precursor for Ach regulates cholinergic neurotransmission in various brain regions. Ach is a neurotransmitter that plays a role in learning, attention and memory (Figure 1).

Cholinergic neurotransmission has been shown to be dysregulated in many neurodegenerative disorders such as Alzheimer’s disease (AD) and this dysregulation may have contributed to neurocognitive impairments often seen in these patients [67]. Postmortem studies revealed a reduction in the levels of membrane phospholipids in aging and demented human brains [68]. It has been suggested that changes in choline-containing phospholipids such as PC could be linked to neuronal membrane breakdown and this may have contributed to the degeneration of cholinergic neurons in Alzheimer’s disease [69]. Whether this loss of membrane phospholipids is a cause or consequence of neurodegeneration is still yet not clear.

## 3. Choline and Other Methyl Donors as Modulators of Neuronal Plasticity Throughout Development

Choline has three methyl groups attached to the nitrogen atom of ethanolamine. It indirectly donates its methyl groups and participate in folate-mediated 1C metabolism for the formation of SAM via its oxidation to betaine by choline oxidase. SAM donates methyl groups to DNMTs that catalyze DNA methylation or HMTs that catalyze histone methylation, two key epigenetic mechanisms that modulate gene expression [8] (Figure 1). Several studies have shown that alteration in choline levels in the brain during critical period of development and in adulthood is associated with changes in epigenetic marks of key genes related to learning, memory, behavior and cognitive functions [29]. This suggests that choline nutrition during early life programs brain development and induces long-term effects on brain functioning via an epigenetic mode of action. This also suggests that this micronutrient is a neuroprotectant that could ameliorate brain cognitive functions across the lifespan.

Studies have shown that dietary components interact with our epigenome and could have long-lasting effects [70,71]. In this section, we will present the beneficial effects of the supplementation of choline during early development on brain cognitive functions later in life. We will also summarize some findings that demonstrated the potential use of this micronutrient and other methyl-donors as supplements to delay the decline in cognitive functions with aging and in some neurodegenerative diseases such as Alzheimer’s disease.

Figure 2 provides an overview of the effects of choline and other methyl-donors at different stages of development on the epigenome and on phenotype.

### 3.1. Choline, Other Methyl Donors and the Developing Brain

Dietary components intake during early life could exert long-lasting effects on mental health [70] as the brain during early development is plastic and undergoes rapid growth and differentiation for its normal functioning. One carbon metabolism is influenced by nutritional status [72]. The components of the 1C metabolism and the folate cycle such as choline, betaine, methionine, folate, VitB6 and VitB12 are pivotal for normal neurodevelopment and neurocognitive functions [73]. Some of these components such as choline, betaine, methionine and folate, participate in the formation of the universal methyl-donor SAM that plays a role in methylation pathways to modulate neuronal gene expression [74].

Animal studies have shown that choline supplementation or depletion during sensitive periods of life such as prenatal, perinatal or adolescence has consequences on neuronal activity and brain function of adult offspring later in life. In the context of cognitive functions, prenatal or perinatal choline supplementation improved animals’ performance in memory-related tasks later on in life [46,47,75,76]. Other studies investigated the epigenetic mechanisms of action of choline and other methyl-donors in the brain. Choline has been shown to modulate gene expression and function by epigenetic mechanisms such as changes in DNA methylation, histone methylation or changes in the expression of specific microRNAs [8]. In vitro studies showed that choline depletion altered hippocampal development in fetal mouse brain and diminished cultured neural progenitor cell (NPC) proliferation [77]. This depletion correlated with global and gene-specific methylation changes in mouse fetal hippocampus [78,79]. In vivo studies demonstrated that choline deficiency at embryonic day E17 decreased the expression of histone-modifying enzymes such as G9a histone-methyltransferase and its associated repressive marks H3K9me1 and H3K9me2 in the subventricular zone and the ventricular zone of the hippocampus with no changes in global levels of these histone repressive marks in the whole mouse fetal brain. The induction of these repressive marks during fetal life correlated with a decrease in binding of the repressor element-1 silencing transcription factor (REST) on RE1 site of the calbindin gene (Calb1) promoter, a regulator of synaptic plasticity, learning and memory, and resulted in an increase in its expression in NPC [80].

Other studies conducted in rodents assessed the effects of methyl-donors deficiency on learning and memory. Mice fed with a diet deficient in folate, methionine and choline (FMCD mice) for 3 weeks during the adolescent period showed impairment in specific tests such as novel object recognition and fear extinction [81]. At the molecular level, FMCD mice showed hypermethylation of Gria1 promoter with a decrease in the hippocampal Gria1 gene expression, a gene involved in synaptic plasticity [81]. These results indicate that methyl-donors deficiency during adolescence, a sensitive developmental period, had impaired learning and memory and altered the expression of Gria1 gene in the mouse hippocampus by altering its methylation status.

In the context of histone mark changes or histone methylation changes, iron-deficiency in the fetal-neonatal period induced a change in the methylation of the brain-derived neurotrophic factor (Bdnf) promoter in the rat hippocampus and caused an elevation in the repressive histone marks such as H3K9me3 but a decrease in the activation histone mark H3K4me3 along Bdnf [82]. These changes in histone marks correlated with a decrease in Bdnf expression, a gene involved in hippocampal plasticity. Gestational choline supplementation (from Gestational day GD11-GD18) to iron-deficient rats reversed these observed epigenetic changes along the Bdnf gene [82].

The effects of choline on the symptoms of genetically-related neurological disorders such as Down Syndrome and Rett Syndrome were also studied. A mouse model of Down Syndrome (Ts65Dn) is characterized by depletion of basal forebrain cholinergic neurons and an impairment in functions related to attention, cognition and memory, symptoms often seen in Alzheimer’s disease. Maternal perinatal choline supplementation in this mouse model improved these functions in the adult offspring [83]. Rett Syndrome is another neurodevelopmental disorder that negatively impacts the cholinergic system which is known to regulate cognitive and motor functions. Postnatal maternal choline supplementation in a mouse model of Rett Syndrome (RTT) improved locomotor activity, motor coordination and behavior in the adult offspring [84].

Several human studies demonstrated the beneficial effects of gestational choline or maternal intake of methyl donors on cognitive functions in offspring during childhood. These effects were manifested as improvement in memory, attention and problem solving and lasted till children school age years [85,86,87]. Other studies reported inconclusive results of the effects of maternal choline intake at several stages of childhood on neurodevelopmental outcomes [12]. Wallace 2018 summarized in an excellent review the effects of choline intake in humans on cognitive functions throughout life [12].

### 3.2. Choline, Other Methyl Donors and the Aging Brain 

The quality and quantity of nutrients that we are exposed to could shape our long-term well-being and mental health. Gene-nutrient interaction is quite complex and has been shown to be regulated by epigenetic mechanisms [21,88]. The role of the micronutrient choline and other methyl-donors in the functioning of the mature brain has been studied in animal models and in humans. For example, choline has been shown to exert neuroprotective effects by lessening or delaying symptoms of cognitive impairments and memory decline with aging [89,90,91], symptoms often seen in individuals suffering from neurodegenerative disorders such as Alzheimer’s disease. Could then an individualized nutrition-based approach be adopted early on in life to boost mental health or delay the progression of these debilitating diseases with aging? Here we summarize some of the findings derived from animal and human studies that documented a potential role of methyl-donors in modulating brain cognitive functions, memory and learning with aging and in neurodegenerative diseases such as Alzheimer’s Disease.

Several studies reported an overall decrease in DNA methylation in Alzheimer’s disease brains. This change was associated with hypomethylation of the amyloid precursor protein (APP) gene promoter with an increase in the expression of APP and an increase in beta-amyloid plaques deposition in specific areas of the brain [92]. Beta-amyloid plaques are strong inducers of oxidative stress and inflammation. Other studies did not find changes in the methylation status of the APP gene promoter. Interestingly, in several areas in the cortex, the hippocampus and in the putamen, a significant reduction in SAM levels with an elevation in SAH levels were reported in Alzheimer’s Disease (AD) cases [93]. SAH elevation is known to be a major inhibitor of DNMT activity which could be a factor in causing this state of hypomethylation in the AD brain. Use of a nutritional supplement that includes folate and VitB complex delayed the progressive decline in memory and improved overall performance in specific tasks in AD patients [94,95]. These findings suggest that these methyl donors such as folate and VitB complex, could be used in an individualized manner as supplements in the aging population to lessen the severity or delay AD symptoms.

Choline-containing compounds such as choline alphoscerate or glycerophosphocholine (GPC) and CDP-choline have been used in limited clinical studies and have shown promising neuroprotective effects and a role against age-related dementia [96]. Choline-containing compounds have also shown promising results in animal models. For example, GPC has been shown to induce Ach release in the rat hippocampus and cause positive structural changes in this area of the brain that is associated with attention, learning and memory [97]. Long-term supplementation of CDP-choline in the rat diet has been shown to prevent the development of spatial memory deficit in aged rats [98]. The effects of CDP-choline and other choline-containing compounds on cognitive functions in humans should be further investigated. Long-term dietary choline supplementation (from 2.5 to 10 months) in an APP/PS1 (PS1 = Presenelin1) mouse model of Alzheimer’s disease reduced the accumulation of amyloid plaques and reduced APP processing in the mouse hippocampus. Moreover, dietary choline supplementation reduced the levels of microglial markers activation, reduced the expression of the alpha7 nicotinic acetylcholine receptor (α7nAchR) in these cells which resulted in a decrease in microglia activation. At the behavioral level, long-term dietary choline supplementation improved spatial memory as assessed by Morris water maze in these mice [99]. These results suggest that long-term dietary choline intake mitigated phenotypes associated with Alzheimer’s disease (AD) in this mouse model. The potential use of a diet approach that includes choline in humans to mitigate AD-like pathology should be further investigated.

A recent study demonstrated a transgenerational effect of choline on brain functioning in a mouse model of AD [100]. Folate metabolism is linked to that of choline. The methyl-tetrahydrofolate reductase (MTHFR) does not only convert tetrahydrofolate (Thf) into 5-methyltetrahydrofolate (5Mthf) but also maintains low levels of homocysteine by converting it into methionine, a precursor for SAM [101]. Choline supplementation in a mouse model of AD ameliorated cognitive functions in the first and second generations of mice and reduced the levels of homocysteine in their brain. Elevated levels of homocysteine have been linked to increased risk of AD development [102]. These observed changes in the AD mouse model in response to choline supplementation correlated with alteration in the expression of key genes related to inflammation, histone marks changes and neuronal death [100].

Another study demonstrated both genetic and nutritional contribution of the methyl donor folate to changes in cognitive functions and brain biochemistry [103]. Mice with MTHFR gene variant (Mthfr^+/-^) or “mild Mthfr deficiency” showed impairment in short-term memory function, increased anxiety and disruption in choline metabolism in the cortex and the hippocampus during the first 10 months of life with a decrease in hippocampal *Bdnf* gene expression. Mthfr^+/+^ mice fed with low dietary folate showed a decrease in SAM levels and Ach, alteration in the expression of synaptic markers (such as synpatophysin), Bdnf, epigenetic enzymes (DNMTs, HDACs) and enzymes of 1C metabolism in the cortex or the hippocampus [103]. These findings suggest that the components of 1C metabolism such as choline and folate are essential for normal cognitive functioning.

Limited human studies showed promising results of increasing choline and other methyl-donors intake on improving cognitive functions or some aspects of memory later in life. For example, the use of a combination of several methyl-donors including choline, VitB12, VitB6 and folic acid had promising results in patients with early Alzheimer’s disease (AD). These patients showed improved memory and enhanced cholinergic signaling [104]. The authors argued that the use of this combination of methyl-donors may have provided essential precursors that contributed to the strengthening of membrane integrity and improved cholinergic signaling in the brain of these patients [104].

Similarly, a nutrient supplement “SOUVENAID” that consists of uridine, omega3-fatty acids (DHA) and choline improved memory scores in patients with early AD and resulted in an increase in synapses and neuronal communication in different brain regions [105,106]. It has been stated in these two studies that the combination of these nutrients in this population of patients may have contributed to an increase in brain PC synthesis and positively impacted the structural and functional integrity of presynaptic and postsynaptic membranes [105,106].

Another study aimed at comparing the levels of nutrients such as choline, folate, and homocysteine in the blood and CSF of AD patients and patients with mild cognitive impairments. Alzheimer’s disease (AD) patients had high CSF homocysteine levels but low levels of plasma choline whereas patients with mild cognitive impairments showed low CSF folate and high CSF homocysteine compared to controls [107]. These results indicate that AD patients show reduction in the levels of nutrients involved in phospholipid synthesis such as choline and folate.

Detection of biomarkers of dementia, which is often seen in Alzheimer’s disease patients is now considered an essential tool in preclinical diagnosis or prognosis of this disease [108]. A human study demonstrated a reduction of a panel of ten lipids (eight of these 10 are choline-containing lipids) in the blood of Alzheimer’s disease (AD) patients compared to healthy controls. These ten lipids have been validated to accurately (>90%) predict mild cognitive impairments or AD within 2 to 3 years’ time period [12,109]. These results suggest that the use of biomarkers should be considered and further investigated for the early diagnosis of these diseases for intervention.

Uptake of choline by the brain from circulation decreases significantly with aging [90], which may necessitate a selective and an individualized approach in increasing choline uptake in the elderly population to meet their specific requirements. This reduction of choline uptake by the aging brain would then induce degradation of PC in neuronal membranes as a compensatory mechanism to keep up with brain demand of choline for Ach synthesis [90]. The reduction of choline with aging has been linked to the degeneration of membrane phospholipids in neurodegenerative disorders and loss or decrease in cholinergic neurons activity in affected brain regions [48,49]. Thus, choline supplementation with other methyl-donors in the elderly or with patients at early stages of AD may contribute to the increase of choline uptake by the aging brain and increase choline levels for Ach synthesis. Additional studies are required to explore the potential role of choline and other methyl donors in reducing or delaying the symptoms of cognitive impairments with aging and in neurodegenerative diseases such as Alzheimer’s disease.

Table 1 summarizes the effects of methyl-donors on brain functioning in animal models and in humans.

## 4. Adverse Effects of Inadequate Intake of Methyl Donors

Animal studies showed that excessive or inadequate intake of methyl-donors has adverse effects on neurodevelopment and cognition. For example, prenatal supplementation of high levels of folic acid (10 times the levels of the control group) resulted in structural, behavioral and molecular changes [110]. At embryonic day E17.5, elevation in folic acid resulted in abnormal development of cortical layers in rats. The levels of phosphocholine, choline and its metabolite betaine were significantly reduced in the embryonic liver with a change in Ach levels in the brain [110]. At 3 weeks of life, rats showed reduced size of the hippocampus, altered thickness of the dentate gyrus and impaired short-term memory. At the molecular level, excess supplementation of folic acid altered the expression of Dnmt3a mRNA, a key epigenetic enzyme that catalyzes the de novo methylation reactions and plays a role in brain development and neuronal maturation. This change in Dnmt3a expression was detected in both the cortex and in the hippocampus with no changes in mRNA expression of Dnmt1, a maintenance methyltransferase, and Dnmt3b, another de novo methyltransferase, in the cortex [110].

Another animal study demonstrated the adverse effects of excess intake of methyl-donors during adolescence [111]. Adolescence is a vulnerable developmental stage since the neural circuitry is not fully mature in some brain regions such as the hippocampus and the prefrontal cortex, brain areas that play a role in memory, executive functioning and cognitive flexibility. Supplementation of excess folic acid (four times the levels of the control group) in rats at postnatal day (PD30–60) resulted in endocrine disruption and memory changes. These rats showed decreased levels of thyroid hormones triiodothyronine T3 and tetraiodothyronine T4 in the periphery with a reduced expression of thyroid hormone receptors (TRα1 and TRα2) in the hippocampus. These endocrine changes in the periphery and in the brain correlated with deficits in learning and spatial memory tasks as demonstrated by the Morris Water Maze [111].

A human study assessed the effects of folic acid supplementation during pregnancy on neurodevelopment in the first year of infant life [112]. Infants of women who took during pregnancy more than the recommended dosage of folic acid supplement showed delayed psychomotor development at first year of life as assessed using the Bayley Scale of Infant Development Test [112]. Readers are directed to an excellent recent review that summarizes the effects of dietary interventions in humans on cognitive functions [113].

These selected studies that we summarized above demonstrate that the dose and duration of methyl-donors’ supplementation at critical stages of neurodevelopment are quite important factors to optimize. The current clinical literature and research about this is limited and not conclusive and there are still challenges to face about the ramifications of nutrients supplementation and in particular methyl-donors supplementation on neurodevelopment and cognitive functions. Methyl donors affects the availability of SAM and modulate epigenetic mechanisms such as DNA methylation or histone methylation and impact neuronal gene expression and activity. The effects of supplementation of these nutrients should be further investigated in animals and in humans to ensure the safety of their intake at different stages of life to prevent undesired outcomes. Optimal and safe dosage of nutrient intake should then be determined at different stages of development and in neurodegenerative diseases. Public adherence to dietary recommendations for nutrients supplementation or intake such as methyl-donor supplementation should then be strongly emphasized and better communicated to patients by physicians and nutritional agencies to prevent or mitigate the adverse effects of inadequate nutrient intake on mental health. Educational programs should then be strongly supported to increase public and health professional awareness about the food sources and the beneficial effects of choline and other methyl donors in health across the lifespan. Choline importance in health and the necessity of increasing awareness about its intake in several populations was discussed at the 2018 Choline Science Summit. We refer the readers to an excellent current review about the outcomes of this Summit discussions for further information [32].

## 5. Conclusions

Choline and other methyl-donors have been shown to have a role in normal neurodevelopment and neurocognitive functions. Choline is now considered a neuroprotectant that modulates expression of key genes related to memory, learning and cognitive functions by epigenetic mechanisms. This micronutrient is not only critical for normal growth and functioning of the early brain, but also may have promising effects in boosting brain functions and delaying or mitigating the decline in cognitive functions with aging and in neurodegenerative diseases such as Alzheimer’s disease. Inadequate intake of methyl donors has also been shown to have adverse effects on neurodevelopment and cognition. The effects of choline and other methyl donors on brain functioning and the optimal dose for intake should be further investigated across all age ranges. Understanding the role of nutrients on mental health and how they interact with our genes throughout life are quite important. This may facilitate the adoption of a nutrition-based individualized approach that includes adequate methyl-donors supplementation early in life to achieve improved cognitive functions and better mental health later in life. This approach in combination with therapeutic drugs should also be carefully studied in the elderly as a viable option to mitigate the adverse effects of aging or neurodegenerative diseases on cognitive functions.

## Figures and Tables

**Figure 1 nutrients-11-02995-f001:**
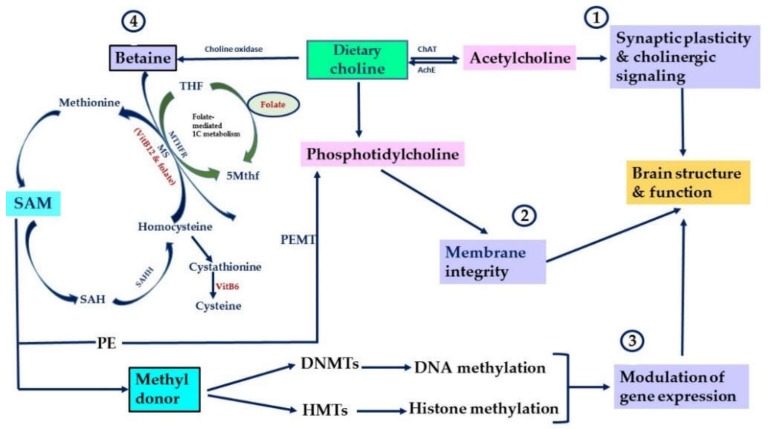
Key physiological functions of choline and its metabolites. Summarizes key physiological functions of choline and its metabolites in the brain. ① Choline is converted via choline acetyltransferase (ChAT) to Ach. ② Choline is converted via several steps into phosphatidylcholine. ③ Choline is converted to betaine via choline oxidase. ④ Betaine contributes to the formation of SAM, main methyl-donor for DNA methyltransferases (DNMTs) and histone methyltransferases (HMTs). Via betaine, choline participates in folate-mediated 1C metabolism. After donating its methyl group, SAM is converted into S-adenosylhomocysteine (SAH), an inhibitor of DNMTs. SAH is hydrolyzed to homocysteine by S-adenosylhomocysteine hydrolase (SAHH). Homocysteine can be converted back into methionine via the transfer of methyl group from 5-methyltetrahydrofolae (5 Mthf) by methionine synthase (MS). MS requires VitB12 and folate for its activity (Modified from [9]).

**Figure 2 nutrients-11-02995-f002:**
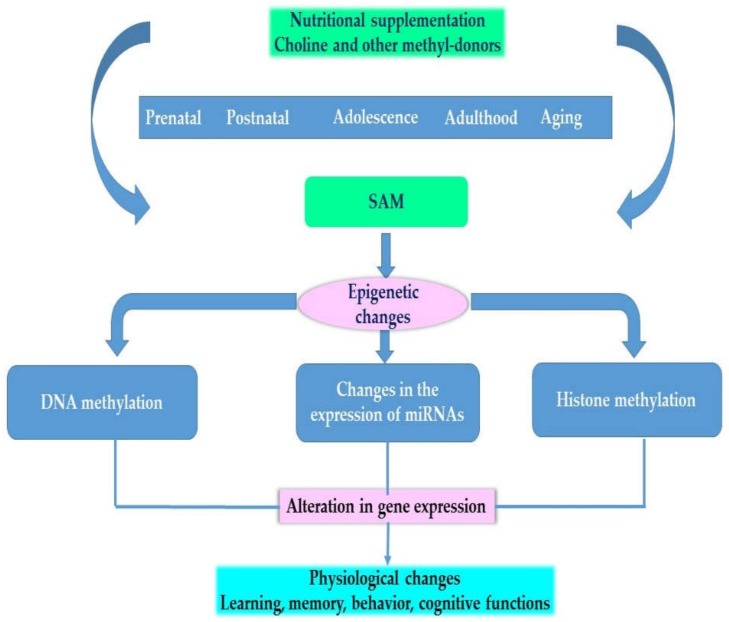
Physiological effects of choline and other methyl donors.

**Table 1 nutrients-11-02995-t001:** Outcomes of methyl-donor supplementation or deficiency on the developing and aging brain.

Description	Outcomes	References
Prenatal and postnatal choline supplementation	Improved memory-related tasks in offspring	[46,47,75,76]
Deficiency of folate, methionine and choline (FMCD mice)	Impaired learning and memory Hypermethylation of Gria1 promoter	[81]
Gestational choline supplementation in iron-deficient (ID) rat	Alteration in hippocampal Bdnf promoter methylation	[82]
Perinatal choline supplementation in a mouse model of Down syndrome	Improved cognitive functions in offspring	[83]
Long-term dietary supplementation of choline in an APP/PS1 mouse model of AD Choline supplementation in a mouse model of AD	Reduced accumulation of amyloid plaques in Reduced α7nAchR expression Reduced microglia activation Improved spatial memory Improved cognitive functions and reduced levels of homocysteine across generations	[99,100]
Low dietary folate in Mthfr^+/+^ mice	Altered expression of synaptic markers, Bdnf and epigenetic enzymes	[103]
Supplementation of choline, VitB12, VitB6 and folate in early AD patients	Improved memory and enhanced synaptic signaling	[104]
Supplementation of SOUVENAID in early AD patients	Improved memory scores	[105,106]

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
