# Peer review of "Neuroprotective Effects of Choline and Other Methyl Donors"

_nutrients, 2019, doi:10.3390/nu11122995_

Round 1

Reviewer 1 Report

I don't have further comment.

A suggestion: Author should have provided a track change version, which would be easier to see the updates.

Author Response

Thank you very much for reviewing my manuscript.  I submitted a revised version with track changes.

Reviewer 2 Report

The author presents a narrative review of methyl donors, with a strong focus on choline.

Could a justification for the addition to the literature be made in light of the following 2 recent reviews?

Wallace, T. C. A Comprehensive Review of Eggs, Choline, and Lutein on Cognition 496 Across the Life-Span. J Am Coll Nutr 2018, 37 (4), 269–285. 497

Wallace, T. C.; Blusztajn, J. K.; Caudill, M. A.; Klatt, K. C.; Natker, E.; Zeisel, S. 499 H.; Zelman, K. M. Choline: The Underconsumed and Underappreciated Essential 500 Nutrient. Nutr Today 2018, 53 (6), 240–253

Occasional grammatical errors and typos could be corrected.

Author Response

Thank you very much for reviewing my manuscript and for pointing me to these two recent reviews.  I included useful information from these two reviews and from other studies listed in these reviews. 

 Please refer to my revised version with track changes to lines: 43-44, 76-78, 96-105, 240-246, 332-338, 399-405.

Round 2

Reviewer 2 Report

the authors did not address the original comments. They lack any rationale for the need for their review or what they add-there are 2 recent reviews that are superior and that cover everything in this review.

This manuscript is a resubmission of an earlier submission. The following is a list of the peer review reports and author responses from that submission.

Round 1

Reviewer 1 Report

Bekdash thoroughly reviewed recent literature on neuroprotective roles of the micronutrient choline and other methyl donors. This review is quite timely, and well written, and will be a contribution to the literature. I don’t have major comment, expect some minor ones.

Line 180: typo “synaptic plasticity”. Line 182: typo “methionine” Line 199: “syndrome” Line 205: “locomotor”

Author Response

Thank you for the comments.

I corrected all these typos in lines 187, 191, 205 & 211.

Reviewer 2 Report

In this manuscript Rola Bekdash describes the potential neurodevelopmental role, and neuroprotective effects, of methyl donors. Overall, the manuscript is well written (with respect to english use and clarity) and is adequately concise. Without being intimately embedded in this literature, references appear appropriate at face-value. However, I do have some broad-strokes recommendations that I believe will improve the manuscript. I implore the author to consider these points not as additional work, but as corollary themes which will benefit not this written body but also future works which evaluate its contents. 

First, references are sometimes not used until the end of several sentences describing the outcomes of studies. I believe the references should be used more frequently, so that the work being described and discussed is easier to locate.

Second, the author makes a number of statements which come across sometimes as vague and other times as repetitive. I would implore the author to revisit major sections and try to integrate more biological mechanisms. These should not be speculative, so not to mislead a naive reader, but should reflect what is known in a greater depth where permitted. For example, molecular changes are rarely discussed, molecular pathways not identified, and behavioral studies vaguely described (e.g. "reaction to fear conditioning" on line 206 is literally useless information - is this an enhancement or deficiency? is it one phase of testing vs another? Is it targeting encoding, retrieval, consolidation, reconsolidation, or something else?). This is but one example. I think the entirety of the manuscript needs a similar level of critical analysis, so that directionality of changes are obvious and that all relevant experimental details are provided so that the result can be interpreted appropriately. 

Third, there is no discussion of evidence against the use or therapeutic benefits of micronutrient diets. This is important, the review surmises positive evidence but it must do so via greater depth of analysis and also acknowledge any evidence that raises questions.

Fourth, and building off my third point, limitations and ethical considerations need to be explicitly considered. This is important. I give the author free reign to consider what to discuss here, but safety, regulation, & potentially deleterious effects should be considered.

Overall, I actually like the article and its conciseness, however these points above will substantially improve it and I will be happy to review these changes in the authors' revision should they be made. 

Author Response

Thank you for the reviews of my paper (nutrients-614417). The reviews were very useful to me in making revisions to the manuscript, and I hope that it will now be suitable for publication.  Please find below my reply in bold. 

Reviewer # 2:
Q1: First, references are sometimes not used until the end of several sentences describing the outcomes of studies. I believe the references should be used more frequently, so that the work being described and discussed is easier to locate.

 I added references, as suggested by Reviewer 2, on lines 189, 198, 226, 236, 248, 253 & 257.

Q2: Second, the author makes a number of statements which come across sometimes as vague and other times as repetitive. I would implore the author to revisit major sections and try to integrate more biological mechanisms. These should not be speculative, so not to mislead a naive reader, but should reflect what is known in a greater depth where permitted. For example, molecular changes are rarely discussed, molecular pathways not identified, and behavioral studies vaguely described (e.g. "reaction to fear conditioning" on line 206 is literally useless information - is this an enhancement or deficiency? is it one phase of testing vs another? Is it targeting encoding, retrieval, consolidation, reconsolidation, or something else?). This is but one example. I think the entirety of the manuscript needs a similar level of critical analysis, so that directionality of changes are obvious and that all relevant experimental details are provided so that the result can be interpreted appropriately. 

The statement on line 212 “but did not improve offspring reaction to fear conditioning” is now deleted.

This manuscript is meant to be a “communication” that summarizes literature review about the neuroprotective effects of methyl-donors across the lifespan.   My intention was not to cover in detail the molecular changes described in several studies.  This is a future manuscript that I am planning to write.

However, I discussed the molecular mechanism in some studies from line 171-212.

Q3: Third, there is no discussion of evidence against the use or therapeutic benefits of micronutrient diets. This is important, the review surmises positive evidence but it must do so via greater depth of analysis and also acknowledge any evidence that raises questions.

Q4: Fourth, and building off my third point, limitations and ethical considerations need to be explicitly considered. This is important. I give the author free reign to consider what to discuss here, but safety, regulation, & potentially deleterious effects should be considered.

I added a section entitled “Adverse Effects of Methyl Donors Supplementation” starting on line 273 to address Reviewer 2 comments (Q3 & Q4).   I also included a summary of these findings at the end of Table 1.